# Phenotypic Characterization and RT-qPCR Analysis of Flower Development in F_1_ Transgenics of Chrysanthemum × grandiflorum

**DOI:** 10.3390/plants10081681

**Published:** 2021-08-16

**Authors:** Saba Haider, Muhammad Ajmal Bashir, Umer Habib, Yike Gao, Muhammad Rashid Shaheen, Rashid Hussain, Fan Min

**Affiliations:** 1National Flower Engineering Research Centre, Beijing Key Laboratory of Ornamental Plants Germplasm Innovation and Molecular Breeding, College of Landscape Architecture, Beijing Forestry University, Beijing 100083, China; sabahaider@bjfu.edu.cn (S.H.); qffanmin@163.com (F.M.); 2Laboratory of Plant Biotechnology and In Vitro Tissue Culture, Department of Agriculture and Forest Sciences (DAFNE), University of Tuscia, 01100 Viterbo, Italy; muhammadajmal@unitus.it; 3Department of Horticulture, PMAS-Arid Agriculture University, Rawalpindi 46300, Pakistan; umer@uaar.edu.pk; 4Department of Horticultural Sciences, Faculty of Agriculture and Environment, The Islamia University of Bahawalpur, Punjab 63100, Pakistan; rashid.shaheen@iub.edu.pk (M.R.S.); rashid.hussain@iub.edu.pk (R.H.); 5Cut-Flower and Vegetable Production Research and Training Cell, The Islamia University of Bahawalpur, Punjab 63100, Pakistan

**Keywords:** RNA silencing, cross breeding, hybrid identification, quantitative real time PCR

## Abstract

Gene silencing is the epigenetic regulation of any gene in order to prevent gene expression at the transcription or translation levels. Among various gene silencing techniques, RNA silencing (RNAi) is notable gene regulation technique that involves sequence-specific targeting and RNA degradation. However, the effectiveness of transgene-induced RNAi in F_1_ generation of chrysanthemum has not been studied yet. In the current study, we used RNAi-constructed *CmTFL1* (white-flowered) and *CmSVP* overexpressed (yellow flowered) transgenic plants of previously conducted two studies for our experiment. Cross hybridization was performed between these intergeneric transgenic and non-transgenic plants of the winter-growing chrysanthemum selection “37” (light pink flowered). The transgene *CmSVP* was confirmed in F_1_ hybrids by RT-PCR analysis, whereas hybrids of *CmTFL1* parental plants were non-transgenic. Besides this, quantitative real-time PCR (qPCR) was used to explain the molecular mechanism of flower development using reference genes. Intergeneric and interspecific hybrids produced different colored flowers unlike their respective parents. These results suggest that generic traits of *CmSVP* overexpressed plants can be transferred into F_1_ generations when crossed with mutant plants. This study will aid in understanding the breeding phenomenon among intergeneric hybrids of chrysanthemum plants at an in vivo level, and such transgenics will also be more suitable for sustainable flower yield under a low-light production system.

## 1. Introduction

Chrysanthemum (*Chrysanthemum ×*
*grandiflorum* Tzvelve.) is a well-known flower originating from China and other Asian countries (Korea, Japan). Chrysanthemum is the second most important floriculture crop in the worldwide floriculture trade [1,2], sharing 30% of the total cut flower production in the world. This flower is an important ornamental crop worldwide with the highest agronomic value, and it also plays an important role in landscaping. After its distribution to all parts of the world, chrysanthemum is being widely planted and cultivated, hence, becoming one of the most famous cut flowers with the most variety among ornamental plants. For flower gardening, the abundance of phenotypic variation in chrysanthemums is the highest among the world’s ornamental crops [3,4]. Its diversity with respect to flower type and color and plant architecture gives it high production value and a large economic market, particularly in the Asian and European flower industries [5]. 

Chrysanthemum (*Chrysanthemum × grandiflorum* Tzvelv.) is one of the perennials belonging to the temperate climatic zone with an optimum temperature range between 18–21° for growth and development [6]. Vegetative development, bud differentiation, and flowering of chrysanthemum could be inhibited at temperatures above 32 °C [7]. Chrysanthemum has approximately 20,000 cultivars in the world, and China has the majority of its cultivars, i.e., about 7000 on record. There is an estimate that more than 90% of their production is through conventional breeding. Additionally, several new chrysanthemum species are being produced through this method annually [8,9]. Despite the fact that various chrysanthemum cultivars are being produced by conventional hybridization techniques, the species is an allohexaploid and self-incompatible (with 3 S alleles) [10,11,12,13,14].

The origin of novel potted and ornamental chrysanthemum plants is from interspecific crosses between wild natives of China, which is considered as the epicenter of the genus *Chrysanthemum* [15]. Crossing and doubling between *Chrysanthemum zawadskii* var. latilobum (Maxim.) Kitamura (2*n* = 2*x* = 18) and *C. indicum* var. *procumbense* (Lour.) Kitamura (2*n* = 4*x* = 36) resulted from florist chrysanthemum (2*n* = 6*x* = 54) [16]. Nevertheless, some limitations are still associated with conventional breeding for florist’s Chrysanthemum, which includes: unstable and variable chromosome numbers that form a hexaploid complex with aneuploidy (2*n* = 54 ± 7∼10), even within a genotype; formation of fragmented chromosomes during mitosis; substantial genome size (25 pg DNA/cell); sporophytic self-incompatibility barrier, inbreeding depression, and genetic load [17,18] although fertile seed derived from the screening of progeny derived from intra- or interspecific crosses of compatible combinations have resulted in many new cultivars having been developed [19,20,21,22]. Contemporary cultivars are hexaploids with a loss or gain of several chromosomes that display a self-incompatible trait. Approximately 40% (ca. 18/50) of *Chrysanthemum* species are originated in Japan [23].

The continuous development and innovation of plant molecular biology provide a new way to improve the characteristics of ornamental plants. Novel genetic, biological, and molecular techniques are required to fulfill consumer demands. For this purpose, interspecific hybridization could be a great source in which the plants that belong to two unlike non-compatible species are crossed. Useful or favorable traits are transferred from wild relatives to cultivated ones or between two different species. Interspecific hybridization has been widely used in the Brassicaceae family to transfer favorable genetic traits to a wide germplasm [24,25]. Many wide hybrids are associated with the expression of negative traits transferred from the donor parent along with the target trait, but these can be eliminated by performing one or more backcrosses with the recipient parent [26]. Commercialization of genetically modified chrysanthemums is associated with the risk of crossing that must be addressed. However, research work on the cross-ability of transgenic plants and the heritability of transgenes in F_1_ hybrids of chrysanthemums is rare [27,28,29]. In this study, the cross-ability of RNAi-induced *CmTFL1* transgenic *Chrysanthemum morifolium* var ‘Fenditan’ and *CmSVP* transgenic overexpressed plants of ‘Ganjue’ cultivar taken from our previously conducted study [30] with one non-mutant selection “37” was examined. The heritability of the transgene (h^2^ = 0.82) and modified physical parameters along with flower color were demonstrated.

## 2. Results

### 2.1. Evaluation of Phenotypic Parameters of Chrysanthemum F1 Hybrids

After exposing hybrids to light at various intensities, the growth and development, photosynthetic and fluorescence parameters of the hybrid plants of all cross combinations were monitored. The hybrids of G × R crosses were relatively very short in height compared to those of other crosses with a maximum height of 42.81 cm at 40% light intensity, whereas hybrids of the cross 37 × G had the maximum height of 51.95 cm at the same light intensity. By decreasing phase-to-light intensity from 100% to 60%, the plant height decreased initially in all F_1_ hybrids, followed by a gradual increase at 40% light strength. Interestingly, transgenic hybrids of the cross 37 × G and G × R were significantly shorter at 60% light irradiance (37.72 cm and 38.82 cm, respectively) than at 40% (51.95 cm and 42.81 cm, respectively), thus, representing a much more important difference within the hybrid set (Figure 1a; Appendix A). Similarly, the diameter of the main stem decreased at 20% light strength, with the least diameter of 0.35 cm for G × R hybrids (Figure 1b; Appendix A). The treatment was significantly different from the control. Leaf length and leaf width were increased from 100% to 40% of light intensity, which started reducing at 20% for all hybrids. The difference between control and treatments D and E was significant in all combinations. The results show that with the decrease in light intensity, the shape of chrysanthemum plants changed, which led towards an increased light use efficiency of the plants with better adaption to the lower light environment.

Photosynthetic parameters were investigated at various light phase intensities for F_1_ hybrids. Among the photosynthetic parameters, the hybrids of 37 × G cross showed the maximum value of light compensation point (LCP) as 60.01 μmol·m^−2^s^−^^1^ at 100% light intensity. Besides, it was found that with the decrease in light intensity from 100% to 20%, the light compensation point (LCP) of chrysanthemum hybrid plants was also reduced to the minimum value of 31.43 μmol·m^−2^s^−1^ by G × R hybrids, and the difference between treatments and control was significant (Figure 2a; Appendix A). On the other hand, the surface quantum efficiency (AQY), net photosynthetic rate (P_n_), stomatal conductivity (G_s_), intercellular CO_2_ concentration (Ci), and transpiration rate (T_r_) were increased initially from treatment A to C, followed by a gradual decrease in E in all hybrids, indicating that the photosynthetic function in leaf was enhanced with a sudden decline afterwards (Figure 2b–f; Appendix A). Conversely, the stomatal limit value (L_s_) was increased rapidly from treatment A to D in all F_1_ hybrids (Figure 2g; Appendix A).

In general, the effect of light intensity on the fluorescence parameters of chrysanthemum hybrid leaves decreased significantly with the relative decrease in light intensity. Fv/Fm ratio for the hybrids of 37 × G gradually increased up to the maximum value of 0.869 at 80% light intensity, which was significantly different from the control, whereas the minimum value was exhibited by the hybrids of G × R parental plants was 0.804 at 100% light strength (Figure 3a; Appendix A). Besides, ΦPSII, F_v’_/F_m’,_ and ETR were compared for treatments B and C, which showed a gradual increasing trend in F_1_ hybrids Figure 3b–d; Appendix A). In addition to the significant difference between ETR and control, the difference between the rest of the parameters was not significant (Figure 3d; Appendix A). When the light intensity was reduced to 40%–20% in treatments D and E, the values of some parameters decreased speedily. Specifically, ΦPSII and ETR were decreased to the minimum value of 0.549 and 87.31 in G × R hybrids, respectively, which were significantly different from the control. It is stated that the weak light strength can stimulate the light energy utilization potential of leaves of hybrid plants, but when the low light reaches a certain degree, it may have a certain effect on the normal function of PSII. In addition, with the decrease in light intensity, an increasing trend was observed in qP, which reduced rapidly afterwards to 0.712 at 20% of light intensity (Figure 3e; Appendix A). Conversely, the value of NPQ decreased, followed by a gradual rise to 2.437, indicating that at a relative light intensity of 80% and 60%, the light energy used for photosynthetic electron transmission increases in PSII, while the light energy consumed in the form of heat dissipation decreases (Figure 3f; Appendix A). However, at a relative light strength of 40% and 20%, PSII consumes more light energy in the form of thermal dissipation, while the light energy used for photo electronic transmission decreases. Due to intra-hybrid comparison, it was found that F_1_ hybrids of 37 × G cross perform better in all morphological parameters than those of G × R and 37 × R.

With respect to all traits, some hybrids resembled one or the other of two parents. All parameters, including plant height, leaf length, leaf width, inflorescence diameter, and time of flowering of all hybrids, were distinguishable from each other and from their respective parents. Besides, the most significant parameter that made them highly distinguishable from each other and parents was flower color. 37 × R hybrids produced purple flowers, while the flower color of R was white and that of 37 light purple; the flowers of G × R hybrids were yellow-colored (R: white and G: yellow); 37 × G produced indigo flowers (G: yellow and 37: light purple) (Figure 4a–c).

### 2.2. Identification of the Transgene CmSVP in F_1_ Hybrids

PCR analysis was performed to amplify a segment of the *CmSVP* gene with the 700 bp target sequence. Among the progeny of G × R transformants, the transgene was detected in five of six strains that were obtained by crossing *CmSVP* overexpressed plants (G) as a seed parent, while among the progeny of 37 × G, the transgene was detected in four of five strains that were obtained by crossing *CmSVP* overexpressed plants (G) as pollen parent as shown in Figure 5. As *CmTFL1* gene expression was interfered in RNAi mediated plants, so F_1_ hybrids of 37 × R were non-transgenic. To check DNA quality, the endogenous chrysanthemum actin gene was amplified and confirmed for every *CmSVP* gene-negative plant. These results verified the presence of a transgene in some of the progeny and further demonstrated that the PCR-positive strains were F_1_ hybrids when one of the parents was transgenic.

## 3. Discussion

Shading profoundly affects plant development, particularly, characters such as height, internode length, shoot diameter, leaf length and width, color, and shape [31,32,33,34,35]. Leaf area varies as a consequence of differences in cell mass and cell wall thickness during a plant’s ontogeny, and its habitat promotes the development of these characteristics in the later stages of its life cycle and its growth environment [36]. The core interest of the study was to explore plant behavior growing under different shading regimes. Observations revealed that shading with 40% and 20% irradiance increased the leaf area of chrysanthemum, and the leaf area of the plants under shade was higher than plants under full sun. Significant evidence has been reported in previous studies that low irradiance induces a shift in plant biomass distribution resulting in enlarged leaf area [32]. Shading various levels of irradiance increased leaf area in jasmine [37] and *Tetrastigma hemsleyanum* [33]. Species such as *Dieffenbachia longispatha*, well adapted to shade, display a modest capacity for photosynthetic acclimation to increase leaf area [38]. 

Leaves are the main organs of plants for the production of photosynthetic assimilates. Impacts of environmental factors on plants or plants’ adaptability to the environment are reflected by the structural attributes of leaves [39]. The interception of Photosynthetically active radiation (PAR) by the plant largely depends on leaf area, which tends to alter during plant growth and development. This relationship is quite significant, being correlated to the photosynthetic efficiency and biomass production in plants [40,41,42]. It is an established fact that an increased leaf area enhances photosynthetic efficiency of plants as reported in Chrysanthemum [43,44,45], Rose [42], Tulip [46], Gladiolus [47], and Eustoma [48].

The biological significance of photosynthesis is to fix carbon and water to synthesize carbohydrates and produce oxygen. Energy derived from photo-assimilates is used for cell division and cell enlargement. It is a well-established fact that reduced photosynthetic activity due to low light intensity hinders plant growth [49,50].

The light compensation point of chrysanthemum hybrids decreased in moderate shade conditions. This indicates that the lower limit of the intensity requirement of light intensity is reduced, and by increasing the surface quantum efficiency to adapt to the reduction of external light intensity, the weak light has certain adaptability [41]. When the light strength drops below 40% of natural light, the surface quantum efficiency begins to decrease, indicating that the photolyase function of the photolysis mechanism is no longer able to adapt to the low-light environment. Many studies have shown that shade usually reduces the net photosynthetic rate of a single leaf [49,51,52]. The Pn value in F_1_ hybrids was found the highest plants exposed to 80% irradiance (Figure 2c; Appendix A). The net photosynthetic rate of chrysanthemum F_1_ hybrids also showed a gradual downward trend under different shade treatments, and the pore conductivity and intercellular CO_2_ concentration were also reduced. The markedly reduced CO_2_-assimilation rate that was recorded for the plants exposed to both full sunlight and 20% irradiance, which occurred due to nonstomatal limitations as the intercellular CO_2_ concentration in the leaves was not lower than that of 80% irradiance-treated plants. At the same time, the value of gs increased significantly in the plants grown under full sunlight (Figure 2e; Appendix A). gs and E values reduced under full irradiance because light saturation implied the plants to close down the stomata to decrease water loss [33], which was not in conformity with the results of the present study. According to Farqhar and Sharkey, the decrease in blade Pn is accompanied by a decrease in Ci and an increase in Ls. The main limiting factor for the photosynthesis mechanism is the pore factor. This is not the same [53] as the results of research by Joe Xinrong and others.

The results of this experiment show that with the decrease in light intensity, the partial closure of the pores of all chrysanthemum hybrid leaves may be the main reason for the decrease in the photosynthetic rate of leaves, while the decrease in non-porous factors (the photosynthetic activity of leaf cells) is a secondary factor [54]. At the same time, the influence of light strength on plant growth is very complex, and shade not only affects light but also affects the concentration of carbon dioxide, temperature, humidity, and other microenvironments, thus affecting the opening and opening of pores; therefore, the decrease in the photosynthetic rate is the result of the combined use of the above factors. 

The plants subjected to high irradiance stress typically show lower Fv/Fm values than those of non-stressed plants [55]. Fv/Fm of leaves in the plants grown under full sunlight fell below 0.8, a sign that photoinhibition occurred [47,48]. In this experiment, the plants grown under full sunlight exhibited lower Fv/Fm than those shaded ones (Figure 3a; Appendix A). The Fv/Fm value changed slightly under 55%, 25%, and 15% irradiance, probably because that the photosynthetic reaction center works well in a certain range of light intensity. 

It was reported that the Fv/Fm value of high-irradiance stressed plants is comparatively less than non-stressed plants [36]. Photo-inhibition was observed in the plants grown under full sunlight with fewer values of Fv/Fm below 0.8 [56,57], which were in conformity with the results of this experiment. The efficacy of the photosynthetic reaction center is largely dependent on the range of light intensity that critically affects the value of Fv/Fm. The results of these experiments were confirmed in Jasmine [37]. The absorption of light energy in chlorophyll molecules can be used in photosynthesis, dissipation as heat or re-emission as chlorophyll fluorescence [55,56,57,58]. The fluorescence parameter fv/Fm of medicinal white chrysanthemum leaves increases with the decline in light intensity, while ɸPSII, Fv’/Fm, ETR, and qP rise under varying light intensities of 100% to 60%. Conversely, NPQ decreases in turn, indicating that the photochemical capacity of PSII tends to rise in a certain low light range and that the absorbed light energy can be more evenly distributed to photochemical pathways. However, when the light strength is less than 40%, ɸPSII parameters such as NPQ begin to decrease, and NPQ rises. Figure 3b shows that the photolytic activity of PSII. The secondary factor that causes the decrease in photochromatic rate, is not unaffected, which is similar to the results [59] of the study of strawberries by Chi Wei and so on and may also be related to the decrease in photochrome (phytochrome) pigment content. In addition, the PSII reaction center consumes excess light energy absorbed by antenna pigments in the form of heat dissipation to avoid inactive or destructive to the photolyzing mechanism. Therefore, the increase in NPQ is a kind of self-protection mechanism of plants, which has a certain protective effect on the photolyzing mechanism [60].

Heavy shading is useful in the reduction of qP [33], which shows the significant difference in the electron transport rate in PSII. These results were confirmed in the current experiment. On the other hand, the NPQ energy is dissipated as heat energy and is not used in the transport of photosynthetic electrons [61,62]. The difference in energy absorption required for photochemical utilization is due to higher NPQ values in the full sunlight-treated plants that might result from photoinhibition. ΦPSII is downregulated as thermal dissipation of excitation energy and functions in PSII centers’ closure [63]. In the current study, the leaves under full sunlight, with 20% and 40% irradiance, showed lower values of ΦPSII than those under 60% and 80% irradiance (Figure 3b; Appendix A). Therefore, it is observed that the former pathway was involved in the reduction of ΦPSII in full sunlight exposure, whereas the latter pathway influenced the reduction of shading.

## 4. Materials and Methods

### 4.1. Plant Materials

The RNAi-mediated *CmTFL1* plants of ‘Fenditan’, *CmSVP*-overexpressed plants of ‘Ganjue’, and non-mutant plants of ‘37′ were grown in the greenhouse of Beijing Forestry University (Beijing, China). The optimal conditions for the photoperiod were maintained along with temperatures ranging between 18 °C and 25 °C.

### 4.2. Hybridization and Seed Setting

Breeding program was organized in accordance with the following 3 crosses:*CmSVP* overexpressed plants (G-♀) × RNAi-mediated *CmTFL1* plants (R♂);Non-mutant *Chrysanthemum* Selection (37♀) × RNAi-mediated *CmTFL1* parent (R♂);Non-mutant *Chrysanthemum* Selection (37♀) × *CmSVP* overexpressed parent (G♂).

*C. morifolium* RNAi-mediated *CmTFL1* parent ‘‘R’’ and *C. grandiflorum CmSVP*-overexpressed parent ‘‘G’’ were subjected to a hybridization experiment with one non-transgenic Chrysanthemum selection (“37”). The tubular florets of G were detached before emasculation, and ligulate flower petals were cut to expose the stigma. To ensure safe crossing, the emasculated inflorescences were enclosed in a paper bag for two days, after which a greater number of pollens from freshly opened flowers of R were transferred to the opened stigma of seed parent G with a brush. After pollination, the flowers were recovered to avoid uncontrolled pollination [64]. The process was performed for all crosses between transgenic and non-transgenic plants. Three crosses were made between the parental plants that were subjected to the seed setting. When the pedicels were subjected to withering (after 60 days of pollination), a total of 60 hybrid seeds were collected from the pollinated inflorescences. Seeds were sown into plastic trays containing a 2:1 (*v*/*v*) mixture of garden soil (1:1:2 mixture of silt: sand: loam) and vermiculite. Trays were kept under artificial growth chambers with controlled light, temperature, and humidity conditions for seeds germination. At the 5-to-6 leaf stage, seedlings were transplanted into medium-sized cups and placed in a greenhouse, followed by transplantation in large pots after one month.

### 4.3. Field Planting of Hybrid Plants

In order to measure the phenotypic characters of hybrid plants more appropriately, a total of 45 hybrid plants were shifted into the open area in Sanqingyuan near Bajia field (Figure 6). The plants were subjected to shade conditions by covering them with a green net for one week to avoid exposure to direct sunlight. Different layers of net sheets were removed slowly to make the plants adapted to sunlight and other environmental conditions more appropriately. Exposure to light intensity was given at 5 treatments of relative light strength, (**A**) 100%, (**B**) 80%, (**C**) 60%, (**D**) 40%, and (**E**) 20%, each treatment with 3 replications. Diurnal variations of light strength quanta were measured using a photo quantum probe of the LI-6400XT portable photosynthetic system (LI-COR^®^ Biosciences, 6400XT, Lincoln, NE, USA) (Figure 7). A total of three plants were taken measured for each treatment, while parameters were measured by taking the fourth functional leaf from top to bottom after one month of transplantation.

### 4.4. Phenotypic Characterization of Hybrid Plants

#### 4.4.1. Measurement of Plant Growth and Development Parameters

Plant growth parameters, including plant height, the diameter of the main stem, leaf length and width of the fourth functional leaf, and leaf length-to-width ratio, were determined for all plants of each treatment in three cross combinations G × R, 37 × R, and 37 × G.

#### 4.4.2. Measurement of Gas Exchange Parameters

The net photosynthetic rate (Pn), intracellular CO_2_ rate (Ci), transpiration rate (Tr), and stomatal conductance (Gs) were measured using an LI-6400XT portable photosynthetic system (LI-COR^®^ Biosciences 6400XT, Lincoln, NE, USA). All gas exchange measurements were recorded at the third fully expanded leaves of chrysanthemum’s F_1_ hybrids were taken from the apex to measure all gas exchange parameters under the conditions of CO_2_ level, PPFD, and air temperature at 400 µmol mol^−1^, 1000 µmol m^−2^ s^−1^, and 25 °C, respectively.

The instrument was used for the measurement of photosynthesis is not a single instrument in itself. It is an assembly of instruments for the measurement of CO_2_, H_2_ O, the temperature of air and leaf, flow rate, PAR, etc. Such parameters were used for calculating transpiration rate, photosynthesis rate, and stomatal conductance. For each treatment of light intensity in all crosses, 5 plants were selected, and 2 leaves were measured per plant on average. The data were analyzed statistically by using SPSS12.0.

#### 4.4.3. Measurement of Chlorophyll Fluorescence 

A chlorophyll fluorometer (PAM–2500, Walz, Effeltrich, Germany) was used to determine chlorophyll fluorescence by taking a second fully expanded leaf from the plant apex at 25 °C. The minimal fluorescence of leaves (F0) was measured at a weak light pulse (< 0.1 µmol m^−2^ s^−1^,600 kHz) for 30 min, whereas and the maximal fluorescence (Fm) induction was performed at saturated light (8000 µmol m^−2^ s^−1^, 20 kHz) for more than 0.8 s. The maximal quantum efficiency of PSII was measured using Fv/Fm, where Fv is the difference between F0 and Fm Fs (steady-state fluorescence yield) and Fm’ (light-adapted maximum fluorescence) were obtained at a light source of 600 µmol m^−2^ s^−1^ for 0.7 s (Fm’ ± Fs)/Fm’ and Fv’/Fm’ were used to calculate the quantum efficiency of PSII (Φ PSII) and the efficiency of the excitation capture by open PSII centers, respectively [65]. Photochemical quenching (qP) was calculated using (Fm’ ± Fs)/(Fm’ ± Fo’) [66]. Non-photochemical quenching (NPQ) was calculated using (Fm—Fm’)/Fm’ [67]. For each treatment of light intensity in all crosses, 5 plants were selected, and 2 leaves were measured per plant on average. The data was analyzed statistically by using SPSS12.0.

#### 4.4.4. Polymerase Chain Reaction (PCR) Analysis

The total DNA was isolated from the leaves of rooted seedlings from three cross combinations by following the protocol of [68]. Two primers of *CmSVP* gene 5′-ATGATGGTTAGGGAGAAAGTGC-3′ (*CmSVP*-F) and 5′-TCAACCTGAGTATGGTAATCCTAAC-3′ (*CmSVP*-R) were used to amplify a 700 bp fragment of SVP gene. A 549 bp fragment of the chrysanthemum *actin* gene was used to evaluate the quality of DNA using two primers 5′- TCCTCTTAACCCAAAGGCCAACAGA-3′ (*CmActin*-F) and 5′-TGAGACACACCATCACCAGAATCCA-3′ (*CmActin*-R). DNA was amplified by using the protocol as mentioned by [69].

#### 4.4.5. Classification of Hybrid Plants

A comparison of the size and shape of leaves and flowers was performed to differentiate F_1_ hybrids from their respective parental plants. Leaf size was measured in terms of length and width. Leaf length was measure from the top to the intersectional point of the leaf and its petiole, whereas width was measured from both ends between the widest part of the leaf perpendicular to the mid-rib by a measuring ruler.

## 5. Conclusions

The above results show that the progenies of the crossed *CmSVP* × RNAi F_1_ hybrids and 37 × *CmSVP* F_1_ hybrids are transgenic chrysanthemum plants and can adapt to mild low-light environments, and when light is less than 40% of natural light, low light affects their normal photosynthesis. Full sunlight is not optimal for various physiological and gaseous parameters of chrysanthemums. Therefore, cultivation and production should adjust the planting density and interproduction density so that the relative light strength control in natural light 80% to 60% is appropriate. Moreover, with the changing patterns and increasing intensity of extreme weather events, sophisticated growing systems are inevitably being well adapted to low light conditions. Therefore, these transgenics can be more successful for low light intensity production systems with enhanced photosynthetic activity and high yield. In our study, we observed a significant increase in leaf area that could be correlated to photosynthetic efficiency and biomass production in chrysanthemum plants. In addition, our current transgenic approach may lay a valuable foundation for breeding strategies such as marker-assisted breeding, gene insertion, and ultimately more sustainable flower production of chrysanthemum.

## Figures and Tables

**Figure 1 plants-10-01681-f001:**
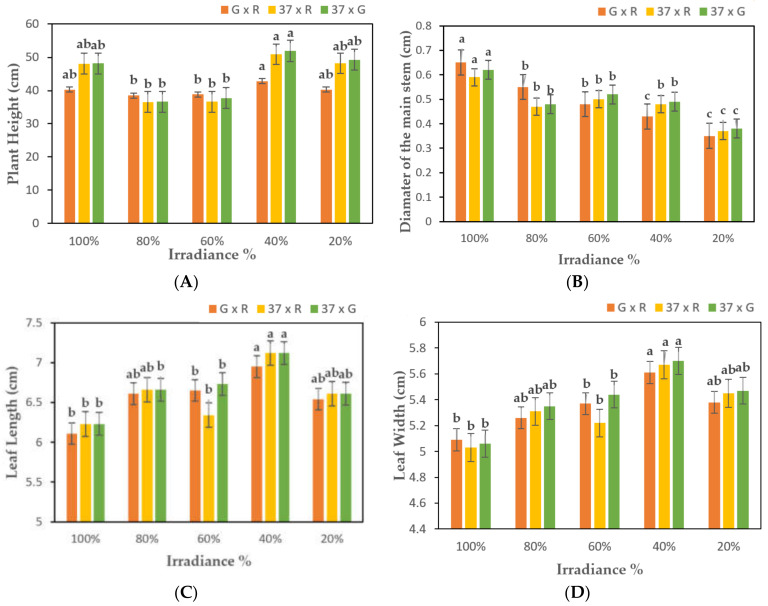
Analysis of growth parameters of F_1_ transgenic and non-transgenic. *Chrysanthemum* plants exposed to 100%, 80%, 60%, 40%, and 20% irradiance; (**A**). Plant Height; (**B**). Diameter of the main stem; (**C**). Leaf Length; (**D**). Leaf Width. G = *CmSVP* overexpressed plants, R = *CmTFL1* RNAi plants, and 37 = Non-mutant selection of *Chrysanthemum.* Three biological replicates were used. Error bars represent ± SE. Values marked by a different letter differ significantly from one another (*p* < 0.05).

**Figure 2 plants-10-01681-f002:**
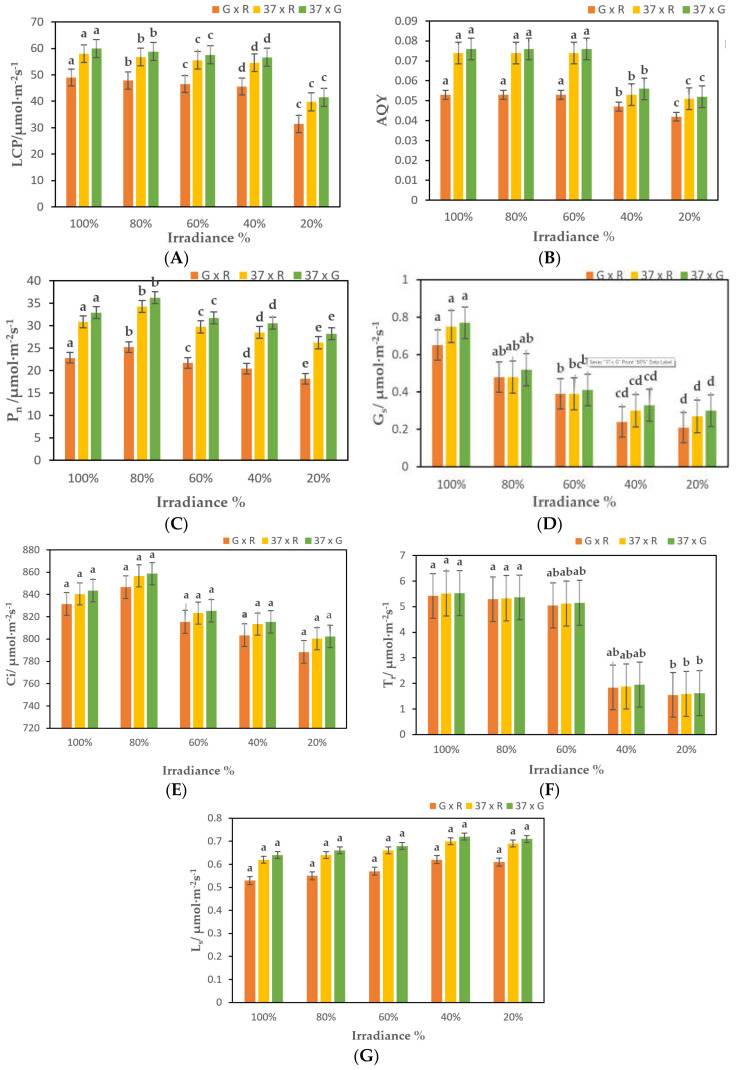
Analysis of leaf gaseous exchange parameters of F_1_ transgenic and non-transgenic *Chrysanthemum* plants exposed to 100%, 80%, 60%, 40%, and 20% irradiance; (**A**). Leaf Compensation Point (LCP); (**B**). Apparent Quantum Yield (AQY); (**C**). Net Photosynthetic Rate (P_n_); (**D**). Stomatal Conductance (G_s_); (**E**). Intra cellular CO_2_ Rate (C_i_); (**F**). Transpiration Rate; (**G**). Stomatal Limit Value (L_s_). G = *CmSVP* overexpressed plants, R = *CmTFL1* RNAi plants, and 37 = Non-mutant selection of *Chrysanthemum.* Three biological replicates were used. Error bars represent ± SE. Values marked by a different letter differ significantly from one another (*p* < 0.05).

**Figure 3 plants-10-01681-f003:**
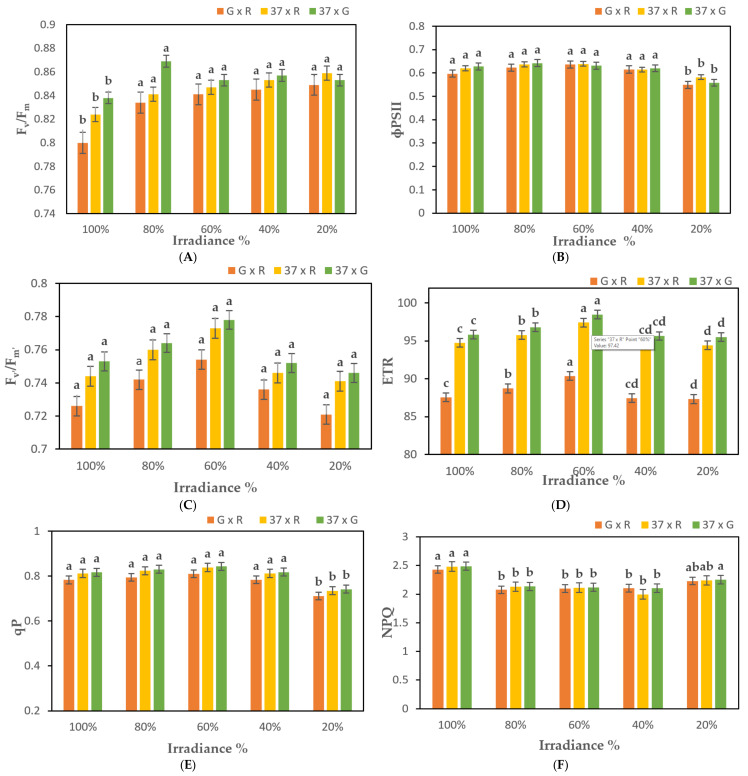
Analysis of chlorophyll fluorescence parameters of F_1_ transgenic and non-transgenic *Chrysanthemum* plants exposed to 100%, 80%, 60%, 40%, and 20% irradiance; (**A**). The Potential Quantum Efficiency of Photosystem II (F_v_/F_m_); (**B**). Effective Quantum Yield (ɸPSII); (**C**). The Quantum Efficiency of open PSII system (F_v’_/F_m’_); (**D**). Electron Transport Rate (ETR); (**E**). The Coefficient of Photochemical Quenching (qP); (**F**). Non-Photochemical Quenching (NPQ). G = *CmSVP* overexpressed plants, R = *CmTFL1* RNAi plants, and 37 = Non-mutant selection of *Chrysanthemum.* Three biological replicates were used. Error bars represent ± SE. Values marked by a different letter differ significantly from one another (*p* < 0.05).

**Figure 4 plants-10-01681-f004:**
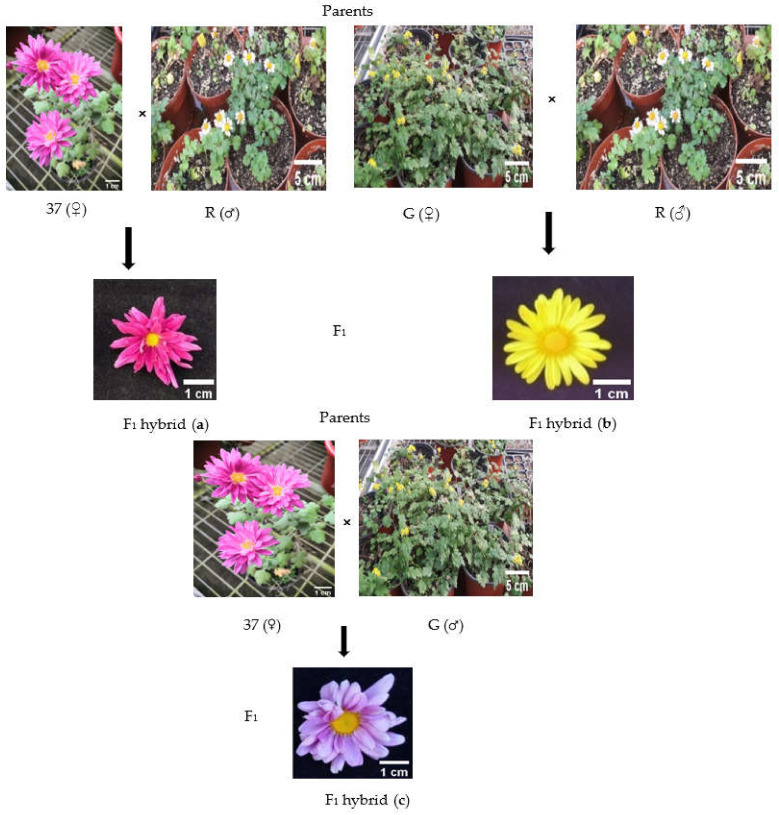
Flower color and inflorescence variegation in RNAi *CmTFL1* (R), *CmSVP* overexpressed (G) and Non-mutant 37 parents and their F_1_ progenies. (**a**) Inflorescence of Non-mutant 37 x RNAi *CmTFL1* (R) parents and their F_1_ progenies; (**b**) inflorescence of *CmSVP* overexpressed (G) x RNAi *CmTFL1* (R) parents and their F_1_ progenies; (**c**) inflorescence of Non-mutant selection37 x *CmSVP* overexpressed parents and their F_1_ progenies. (Photo credit: Saba Haider).

**Figure 5 plants-10-01681-f005:**
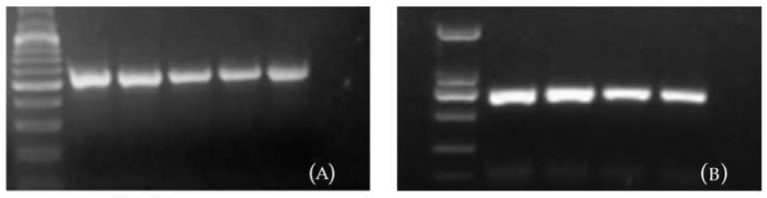
The detection of *CmSVP* in transgenic F_1_ hybrids of chrysanthemum; (**A**). PCR detection of *CmSVP* × RNAi F_1_ hybrids, M: DL2000 Marker; 1–5 indicated transgenic resistant F_1_ seedlings, 6 was plasmid control; (**B**). 37 × *CmSVP* F_1_ hybrids, M: DL1000 Marker; 1–4 indicated transgenic resistant F_1_ seedlings, 5 was plasmid control (Photo credit: Saba Haider).

**Figure 6 plants-10-01681-f006:**
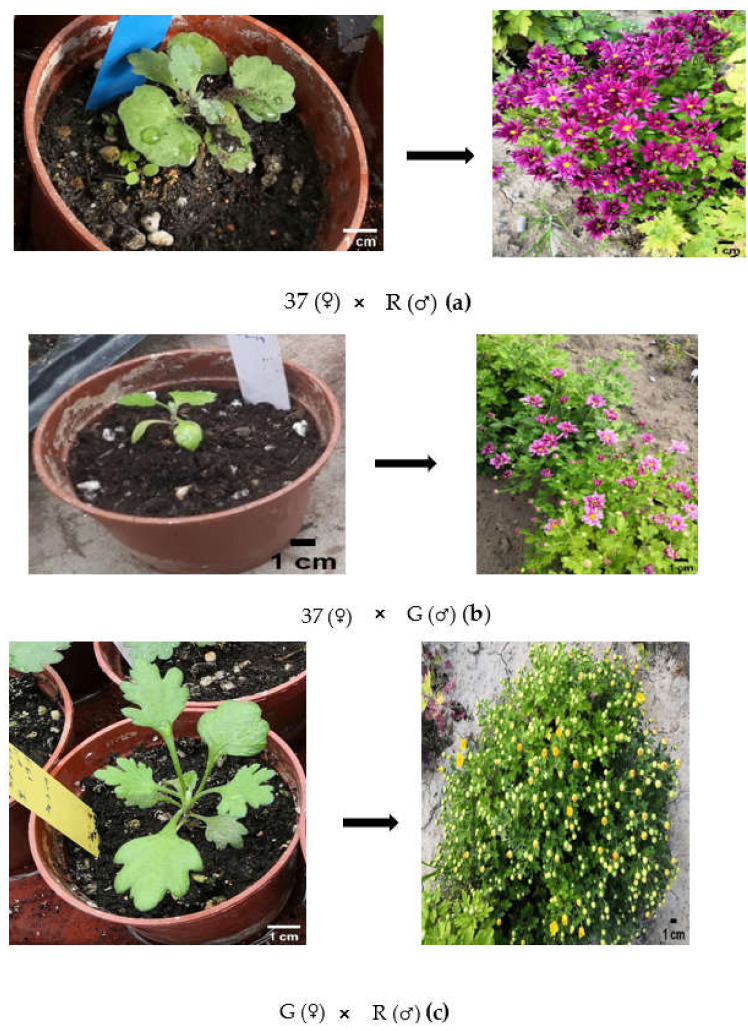
The cultivation of F_1_ transgenic and non-transgenic seedlings of Chrysanthemum from greenhouse to an open field; (**a**)**.** Transplanting of F_1_ non-transgenic seedlings of the cross 37 (♀) × R (♂); (**b**). Transplanting of F_1_ transgenic seedlings of the cross 37 (♀) × G (♂); (**c**). Transplanting of F_1_ transgenic seedlings of the cross G (♀) × R (♂). G = *CmSVP* overexpressed plants, R = *CmTFL1* RNAi plants, and 37 = Non-mutant selection of *Chrysanthemum.* (Photo credit: Saba Haider).

**Figure 7 plants-10-01681-f007:**
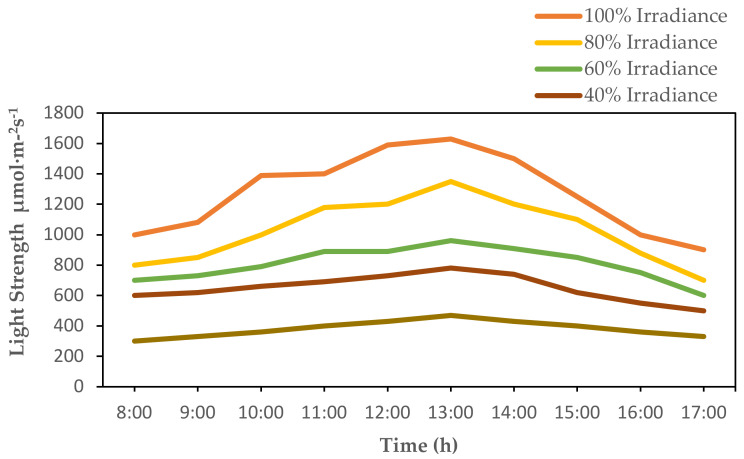
Diurnal variation of light strength quanta under 100%, 80%, 60%, and 20% irradiance (wavelength range 400–700 nm).

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
