# Peer review of "Phenotypic Characterization and RT-qPCR Analysis of Flower Development in F1 Transgenics of Chrysanthemum × grandiflorum"

_plants, 2021, doi:10.3390/plants10081681_

Round 1

Reviewer 1 Report

all of my edits are in the attached Ms (using track changes)

Author Response

Response to Reviewer 1 Comments

Reviewer #1: All of my edits are in the attached MS (using track changes).

Point No A1: Numbered genotypes aren’t cultivars until named.

Response 1: Thank you for the correction. The pointed mistake is corrected in Line # 27 of the manuscript.

Point No A2: Morifolium is no longer the correct specific epithet for this species.  

Response 2: Thank you for the correction. The pointed mistake is corrected in the Line # 39.

Point No A3: Odd statement. It would be better to say that the species is an allohexaploid and self-incompatible (with 3 S alleles); need to increase your citations as well.

Response 3: Thank you for the suggestion. The statement is corrected and citations are added in the Line # 60.

Point No A4: Citations needed.

Response 4: Thank you for the suggestion. The citations are added in the Line # 89.

Point No A5: What was this one?

Response 5: Thank you for the query. The non-mutant selection of Chrysanthemum plants was “37”, as mentioned Materials and Methods and in all figures.

Point No A6: Need specifics on where these were obtained from, their pedigree or ancestry, etc.

Response 6: Thank you for the query.  The RNAi-induced CmTFL1 transgenic Chrysanthemum morifolium var ‘Fenditan’ were taken from the previously conducted study of our research group which will be sent for publication soon, whereas the CmSVP transgenic overexpressed plants of ‘Ganjue’ cultivar were also obtained from one of our previous studies that published few months back. The reference of the published study is as under:

[Yaohui, G.; Pu, W.G.; Bin, M.; Jie, X.F.; Ming, M. Analysis of the Structure and Expression Pattern of Chrysanthemum CmSVP Gene. Henan Agric. Sci. 2021, 50(2), 124-129.]

Point No. A7: I have no idea which of the three parents identified above these codes represent. Very unclear!

Response 7: Thank you for your query. The details of the codes used for parental identification are mentioned in Materials and Methods section in the Lines # 353-355, and also in the captions of the figures 1, 2, 3, 4, and 6.

Point No. A8: Significantly shorter? What do you mean separations tell you?

Response 8: Thank you for your query. The name of the cross was written wrong mistakenly and is corrected in the Line # 99 accordingly. By the sentence, it means that the overall plant height of F1 hybrids of the cross G x R were comparatively shorter than those of other hybrids at all treatments of light irradiance, specifically at 40%. The trend of plant height decreased initially in all F1 hybrids at 100% to 60% light irradiance, followed by a gradual rise at 40%.

Point No. A9:  But you missed the most important difference within one cross 37 x G for height where one treatment (C) was also significantly shorter than treatment D (51.95 cm). A much more important difference within a hybrid set!

Response 9: Thank you for the critical analysis. The similar trend was observed in the hybrid set of G x R cross that shows the different phenotypic behavior of transgenic and non-transgenic hybrids. The results are mentioned in the Lines # 104-107 of the manuscript.

Point No. A10:  No such terms as “weak” and “strong” light! It’s always described in terms of light intensity.

Response 10: Thank you for the correction. The pointed mistake is corrected in the Line # 114.

Point No. A11: Specify in the table what these are!

Response 11: Thank you for the suggestion. The details of the treatments are mentioned in figure 1, 2, and 3

Point No. A12: And where is this data? It should be in Table 1 or in the text so we know whether or not these shade cloths (20%-100%) were real values or not!

Response 12: Thank you for the suggestion. The data for light strength quanta is added in Materials and Methods in figure 7.

Point No. A13: Need bars in each picture for scale. Align the pictures in (b) to not be angled.

Response 13: Thank you for the suggestion. Bars are added in each picture of figure 4 and 6. The angled pictures are replaced in figure 6 of Materials and Methods.

Point No. A14: This is entirely too vague of a figure legend.

Response 14: Thank you for pointing out the mistake. The figure 1 and 2 of Materials and Methods section the previous draft are replaced with figure 6, and the legend/caption is also revised.

 Point No. A15: Also vague. Need photo credits and a bar for scale.

Response 15: Thank you for the suggestion. The legend/caption of a figure 6 is revised, and photo credits and a bar scale is added in figure 4 and 6.

Point No. A16: What were they?

Response 16: Thank you for the query. The three cross combinations were G x R, 37 x R, and 37 x G, which are added in Line # 417-418.

Point No. A17: How was this done?

Response 17: Thank you for the query. Leaf size was measured in terms of length and width. Leaf length was measure from the top to intersectional point of leaf and its petiole, whereas width was measured from both ends between the widest part of the leaf perpendicular to the mid-rib by a measuring ruler. These details are added in Line # 459-462.

Reviewer 2 Report

The authors use RNAi constructed CmTFL1(white-flowered) and CmSVP overexpressed (yellow-flowered) transgenic plants and did the cross-hybridization between these intergeneric transgenic and non-transgenic plants.
   The title needs to change. This work did not just do RNAi but also the overexpressing of the CmSVP gene. But the overexpression work or RNAi work did not confirm by the experiment, such as QRT-PCR. What is genes’ expression, are they really overexpression? Or silencing? The description of the methods part is not clear. How many plants are used for each cultivar? In the pot or in the field experiment? How many replicates?
A lot of mistakes in the references!!!

Author Response

Response to Reviewer 2 Comments

Reviewer #2: The authors use RNAi constructed CmTFL1 (white-flowered) and CmSVP overexpressed (yellow-flowered) transgenic plants and did the cross-hybridization between these intergeneric transgenic and non-transgenic plants.

Point 1: The title needs to change. This work did not just do RNAi but also the overexpressing of the CmSVP gene. But the overexpression work or RNAi work did not confirm by the experiment, such as QRT-PCR. What is genes’ expression, are they really overexpression? Or silencing? The description of the methods part is not clear. How many plants are used for each cultivar? In the pot or in the field experiment? How many replicates?

Response 1: Thank you for the queries. The title of the manuscript is changed. The RNAi work of CmTFL1 and overexpression work of CmSVP were from previous research of our group. Among them, the manuscript on RNAi will be sent for publication soon, while the reference of the study on overexpression of CmSVP gene is stated below:

[Gao Yaohui, Wei Guang Pu, Ma Bin, Xiao Feng Jie, & Ma Ming. (2021). Analysis of the Structure and Expression Pattern of Chrysanthemum CmSVP Gene. Henan Agricultural Sciences, 50(2), 124-129]

For each cross, a total of fifteen plants were used with five treatments of light irradiance in three biological replicates. In Line # 370-375, it’s mentioned that the seedlings were transplanted into medium sized cups and placed in a green house, followed by transplantation in large pots after one month. To measure the phenotypic characters of hybrid plants more appropriately, total 45 hybrid plants were shifted into the open field.

Point 2: A lot of mistakes in the references.

Response 2: Thank you for pointing out the mistakes. The references are revised according to the journal’s template.

Reviewer 3 Report

The authors have presented the findings on understanding the physiology behind Chrysanthemum crosses. The conclusions obtained take us a step forward towards achieving a higher yield in the long run.

For the authors,

I would suggest italicising the scientific names.

I would also suggest making Tables 1, 2 and 3 into graphs considering it is less challenging to recognise patterns. The tables could be a part of the supplementary section.

Author Response

Response to Reviewer 3 Comments

Reviewer #3: The authors have presented the findings on understanding the physiology behind Chrysanthemum crosses. The conclusions obtained take us a step forward towards achieving a higher yield in the long run.

Point 1: For the authors, I would suggest italicizing the scientific names.

Response 1: Thank you for your kind response. The scientific names are checked and italicized in the manuscript.

Point 2: I would also suggest making Tables 1, 2 and 3 into graphs considering it is less challenging to recognize patterns. The tables could be a part of the supplementary section.

Response 1: Thank you for your suggestion. The table 1, 2, and 3 are converted into graphs and added in the manuscript as Figure 1, 2, and 3, while the tables are added in the supplementary section.

Round 2

Reviewer 2 Report

The manuscript is good for publication now.